# Fatty Acid and Amino Acid Profiles of Seven Edible Insects: Focus on Lipid Class Composition and Protein Conversion Factors

**DOI:** 10.3390/foods12224090

**Published:** 2023-11-10

**Authors:** Cristina Perez-Santaescolastica, Ilse de Pril, Ilse van de Voorde, Ilse Fraeye

**Affiliations:** 1Research Group of Meat Technology & Science of Protein-Rich Foods (MTSP), Department of Microbial and Molecular Systems (M2S), KU Leuven—Ghent, Gebroeders De Smetstraat 1, 9000 Ghent, Belgium; ilse.fraeye@kuleuven.be; 2Research Group Enzyme, Fermentation and Brewing Technology (EFBT), Department of Microbial and Molecular Systems (M2S), KU Leuven—Ghent, Gebroeders De Smetstraat 1, 9000 Ghent, Belgium; ilse.depril@kuleuven.be (I.d.P.); ilse.vandevoorde@kuleuven.be (I.v.d.V.)

**Keywords:** *Acheta domesticus*, *Blaptica dubia*, index of atherogenicity, index of thrombogenicity, species-specific conversion factor, amino acid scores

## Abstract

This study is based on the fatty acid and amino acid profiles of seven edible insect species: *Acheta domesticus*, *Alphitobius diaperinus*, *Blaptica dubia*, *Galleria mellonella*, *Locusta migratoria*, *Tenebrio molitor,* and *Zophobas morio*. The aim of the present study is to provide new data on the fatty acid distributions among lipid classes as well as the species-specific protein conversion factor (Kp) of a wide range of insects in order to further improve the nutritional characterisation of insects as food. Oleic acid was the predominant fatty acid in all insects except for *A. domesticus*, in which a significantly higher percentage of linoleic acid was found. The majority of the lipids were neutral lipids. A significant amount of α-linolenic acid in the phospholipid fraction of *L. migratoria* was shown, while in *T. molitor,* phospholipids were the only fraction in which a measurable amount of docosahexaenoic acid was found. Overall, in most insects, the phospholipid fraction had the highest polyunsaturated fatty acid content compared to the other classes, which may be protective in terms of auto-oxidative stability. Kp values in the range of 4.17 to 6.43 were obtained. Within the nutritional quality indices, all insects showed healthy fatty acids and high-quality amino acid profiles.

## 1. Introduction

According to a recent report published by the United Nations, in 2050, the global population will include over 9 billion people [1]. The fast growth of the world population is a great preoccupation since the production of food from animal sources cannot follow this increase [2]. Hence, alternative protein sources are required [3,4]. In this regard, insect rearing has been considered in recent years due to a lower space requirement, the possibility of using organic side streams as feed, and its lower greenhouse gas emissions, which makes it much more environmentally friendly compared to traditional animal production [5]. In addition, insects have a high nutritional value, with protein content ranging, by dry matter, from 13% to 77%, including a high level of essential amino acids (EAA) that could reach, depending on the species, 46–96% of the total amount [6]. The lipid content varies depending on several factors, such as the life stage and feed, from 10% to 60% in dry matter, often showing a high content of linoleic, oleic, and α-linolenic acid [7].

It is estimated that nearly 2000 insect species are consumed by approximately 2 billion people on a regular basis globally, mostly in Africa, Thailand, and Australia [5]. Nonetheless, the practice of eating insects is far less common in European countries. In this regard, the European Regulation (EU) 2015/2283 [8], established on 1 January 2018 by the European Union, was a significant breakthrough in the direction of introducing edible insects in European countries. Through this regulation, edible insects have come to be considered a novel food, allowing them to be commercialised in the European Union upon their approval. Despite the large range of products from insects that are consumed, such as honey or cochineal that is used as a colourant in food manufacture, in Western societies, eating insects is seen by many people as “disgusting”, “unclean”, and “disease-transmitters” due to sanitary, cultural, or even religious reasons, often driven by fear of the unknown [9]. However, appropriate information and the gradual introduction of insects into food products could lead to a change in the mentality of consumers, as has already been observed for other products [10]. 

The amino acid and fatty acid profiles of many edible insects have been studied [11,12,13,14]. In addition, a limited number of more detailed lipidomic studies have been conducted [15,16,17,18]. Nevertheless, the distribution of fatty acids over the respective lipid classes, namely, neutral, glycolipids, and phospholipids, remains unknown. To the best of our knowledge, only Guil-Guerrero et al. [19] reported the fatty acid composition of the lipid classes in a number of insects. However, this distribution is important since it could influence the response to lipid oxidation as well as the bioavailability of essential fatty acids. For instance, although there is no general consensus, there are some studies in which neutral lipids have been observed to undergo auto-oxidation at a faster rate, whereas phospholipids have been observed to be oxidised at a higher level by photo-oxidation [20,21]. It has also been observed that essential fatty acids such as EPA have a higher bioaccessibility in the intestine when they are present in the phospholipid fraction, probably due to a more stable emulsification due to the emulsifying properties of phospholipids [22].

As commented before, together with lipid content, the protein content of various edible insects has been widely reported [23]. However, the protein content reported in the literature is commonly calculated based on the nitrogen content by applying a conversion factor that typically corresponds to 6.25. Due to the presence of non-protein nitrogen, this index may result in an overestimation of the protein content. The use of a correct conversion factor allows a more accurate estimation of protein content, which is essential for further applications in food technology. To date, this species-specific conversion factor has been determined for a limited number of edible insect species, including *Alphitobius diaperinus* (ALD) [24,25], *Acheta domesticus* (ACD) [24,26,27], *Locusta migratoria* (LM) [26], and *Tenebrio molitor* (TM) [24,25,26]. Conversely, there is no direct data on other commonly consumed insects such as *Blaptica dubia* (BD), *Galleria mellonera* (GM), or *Zophobas morio* (ZM).

This work tries to fill these gaps by providing novel data regarding fatty acid distribution among lipid classes, as well as the species-specific protein conversion factor (Kp) of a large range of insects, in an attempt to further contribute to improving the nutritional characterisation of insects as food.

Additionally, the generated fatty acid and amino acid profiles were used to calculate several indices to assess the nutritional quality of lipids and proteins.

Seven insects belonging to different families were selected based on a report of the Scientific Committee of the European Food Safety Authority (EFSA) [28], which describes insects belonging to Orthoptera, Coleoptera, and Lepidoptera as the most commonly known edible insects. The selected insects include *A. domesticus*, *A. diaperinus*, *B. dubia*, *G. mellonella*, *L. migratoria*, *T. molitor,* and *Z. morio*, which belong to the orders Orthoptera, Coleoptera, Blattodea, and Lepidoptera, which cover the worldwide most consumed orders [28].

## 2. Materials and Methods

### 2.1. Edible Insect Samples

All insects used in this study were purchased from a local producer (Nusect Living Nutrition, Deerlijk, Belgium). ZM, TM, ALD, and GM were obtained in larval form, whereas BD, ACD, and LM were collected in their adult stages. After overnight starvation, all insects were freeze-killed by keeping them at −20 °C for 24 h. One fraction of each insect was frozen at −20 °C, while the other fraction was lyophilised, ground, and vacuum stored at −20 °C until further analysis.

### 2.2. Total Lipids and Lipid Classes

Total lipids were extracted from 100 mg of insect powder following the method described by Ryckebosch et al. [29] using chloroform/methanol (1:1), and the total lipid content was determined gravimetrically after the evaporation of solvent by a Laborota 4000 efficient rotavapor (Heidolph, Schwabach, Germany).

From these lipid extracts, the content of different lipid classes (neutral lipids, glycolipids, and phospholipids) was determined following the method described by Christie and Han [30] through separation on a J. T. Baker silica SPE column (Avantor Inc., Radnor, PA, USA). The column was conditioned using two times 5 mL of chloroform, after which approximately 10 mg of lipid, previously dissolved in 100 µL of chloroform, was added to the column. A total of 10 mL of chloroform, followed by 10 mL of acetone, and finally 10 mL of methanol, was used to elute fractions of neutral lipids, glycolipids, and phospholipids, respectively. The content of all fractions was determined gravimetrically after the evaporation of the solvent on a rotavapor. Ten replicates per insect were performed, and after separation, replicates of the same fraction of each insect were pooled and stored at −18 °C until further analyses of the fatty acid composition of each fraction.

### 2.3. Fatty Acid Profile and Nutritional Indices

For fatty acid analysis, 5 mg of lipid from each lipid class was used. Methylation and analysis by gas chromatography with flame ionisation detection (GC/FID) were carried out following the method described by Ryckebosch et al. [29]. For methylation, lipid extracts were dissolved in 1 mL of toluene and 2 mL of sulfuric acid in methanol (1%), keeping the mixtures overnight at 50 °C. After the addition of 5 mL of water with NaCl (5%), the formed fatty acid methyl esters (FAMEs) were extracted with hexane and quantified using GC/FID. An EC_Wax column (Restek, Bellefonte, PA, USA) was used for the analysis with the following temperature-time programme: 70 °C to 180 °C at 10 °C/min, 180 °C to 235 °C at 4 °C/min, and 235 °C during 4.75 min. Peak areas were quantified with Chromcard for Windows software, version 10 (Interscience, Louvain-la-Neuve, Belgium). Standards (Nu-check, Elysian, MN, USA) of 35 different FAMEs were analysed for peak identification. Results were expressed as a percentage of the total FAMEs in each lipid class. The FAMEs of the total lipids were calculated from the FAMEs in the three lipid classes and the corresponding percentage that each class represents.

Total saturated fatty acids (SFA), total monounsaturated fatty acids (MUFA), total polyunsaturated fatty acids (PUFA), total PUFA n − 3 and PUFA n − 6, as well as the PUFA/SFA and n − 6/n − 3 ratios, were calculated. For all insects, the index of atherogenic (IA), index of thrombogenic (IT), and hypocholesterolemic/hypercholesterolemic ratio (h/H) were also determined for the total lipid content according to the following equations [31,32,33,34]:IT = (C14:0 + C16:0 + C18:0)/[(0.5 × ƩMUFA) + (0.5 × Ʃn − 6) + (3 × Ʃn−3) + (n − 3/n − 6)](1)
IA = [C12:0 + (4 × C14:0) + C16:0]/(ƩMUFA + Ʃn − 6 + Ʃn − 3)(2)
h/H = (C18:1 + ƩPUFA)/(C12:0 + C14:0 + C16:0)(3)

### 2.4. Amino Acid Composition and Conversion Factor (Kp)

Amino acids were hydrolysed by adding 25 mL of HCl (6 M) to 50 mg of freshly ground insect and keeping it at 110 °C in an oven for 22 h. Next, derivatisation was carried out with the use of the AccQ•Tag Derivatization Kit (Waters, Milford, MA, USA. Amino acids (with the exception of tryptophan) were determined using an Ultra Performance Liquid Chromatography (UPLC) separation system by Waters (Waters, Milford, MA, USA) with an AccQ•Tag Ultra column (2.1 i.d. × 100 mm from Waters, Milford, MA, USA). The column temperature was 60 °C, the gradient elution was applied according to the AccQ•Tag Ultra method, the flow rate was 0.7 mL/min, and the time of separation was 9.5 min. Three replicates per insect were performed, and data processing was carried out using Empower 2 software (Vesion 2).

After analysing the amino acids, the true protein content was calculated using Equation (1), described by Biancarosa et al. [35]:(4)$$True\;protein\;content=\sum_{i}E_{i}=\sum_{i}\left[AA_{i}\times \left(\frac{AA_{i}\;\left(MW\right)-H_{2}O\left(MW\right)}{AA_{i}\;\left(MW\right)}\right)\right]$$
where $E_{i}$ is the mass of each single amino acid residue in a protein, that is, its true fraction after the loss of one molecule of H_2_O expressed as g/100 g of dry matter (DM), $AA_{i}$ is the mass of each single amino acid expressed as g/100 g of DM, and MW is the molecular weight of each single amino acid.

In order to determine the species-specific conversion factor (Kp), the total nitrogen content in each insect was analysed following the Kjeldahl method, in which an initial digestion stage is undertaken using a K-437 Buchi-Digest System (Buchi, Flawil, Switzerland) coupled to a scrubber B-414 (Buchi, Flawil, Switzerland), followed by distillation through the use of a K-350 Buchi distillation unit (Buchi, Flawil, Switzerland), and finally by titration with a 785 DMP Titrino (Metrohm AG, Herisau, Switzerland). The Kp was calculated following Equation (5) as presented by Mariotti et al. [36], in which $E_{i}$ represents the mass of a single amino acid residue (true protein, see Equation (4)) expressed as g/100 g of DM and TN represents the total nitrogen (g/100 g of DM).
(5)$$Kp=\frac{\sum_{i}E_{i}}{TN}$$

### 2.5. Amino Acid-Based Nutritional Indices

The ratio of essential to total amino acids was calculated as E/T (%), and the essential amino acid index (EAAI) was evaluated by the formula described by Kavle et al. [37], which is based on the content of all EAA except tryptophan in comparison to the reference protein:(6)$$EAAI=\sqrt[{n}]{\sum_{i}\frac{EAA_{i}}{EAA_{is}}}$$
where *n* is the number of EAA, EAAi is the mass of each single EAA expressed as mg/g of protein, and EAAis is the mass of each corresponding EAA in standard expressed as mg/g of protein. The reference substance used was the whole egg, in accordance with Yu et al. [38].

The essential amino acid score (AAS) expressed in percentage was calculated by the FAO/WHO method [39] as shown below:(7)$$AAS=\frac{mg\;amino\;acid\;per\;g\;reference\;protein}{mg\;amino\;acid\;per\;g\;standard\;pattern}\times 100$$

The AAS is the lowest value obtained from the calculations of all amino acids, and the corresponding amino acid is the limiting amino acid.

The predicted protein efficiency ratio (PER) values were calculated from their amino acid composition based on the following equations [40]:(8)$$PER\left(1\right)=-0.684+0.456\left(Leu\right)-0.047\left(Pro\right)$$
(9)$$PER\left(2\right)=-0468+0454\left(Leu\right)-0.105\left(Tyr\right)$$

### 2.6. Statistical Analysis

The values are presented in terms of mean values and the standard error of the means. Differences among insects were examined using a one-way ANOVA, where insect species was set as a factor. When a significant effect (*p* < 0.05) was detected, means were compared using Tukey’s post hoc test. All analyses were conducted using the IBM SPSS Statistics 28.0 programme (IBM Corporation, New York, NY, USA) software package.

## 3. Results and Discussion

### 3.1. Total Lipid Content and Fatty Acid Composition

The highest total lipid content (Appendix A), by far, was found in GM (53.63 g/100 g DM), which was followed by ZM and LM (33.97 and 32.28 g/100 g DM, respectively), TM, ALD, and ACD (22.61, 21.82, and 21.32 g/100 g DM, respectively), and finally BD (13.96 g/100 g DM). The percentage obtained for ACD was comparable with the range described in the literature [41], as was the content of ALD and BD [17]. Likewise, similar values for GM and LM were previously reported [42,43,44]. In contrast, the values observed in TM were slightly lower than the ones reported by Finke [11] and Tzompa-Sosa et al. [17].

The fatty acid profiles of the total lipids of the selected seven edible insects are presented in Table 1. A total of nine fatty acids were detected in amounts above 1% of the total fatty acid content, including three SFAs (C14:0 or myristic acid, C16:0 or palmitic acid, and C18:0 or stearic acid), four MUFAs (C16:1 or palmitoleic acid, C18:1 or oleic acid, C20:1 or 11-eicosanoic acid, and C24:1 or nervonic acid), and two PUFAs (C18:2 or linoleic acid and C18:3n − 3 or α-linolenic acid). Among SFA, a low percentage of myristic acid was present only in LM and TM (1.67 and 3.57% of the total FA, respectively), while palmitic acid and stearic acid were found in all insects [17]. According to previous findings, this can be explained by the de novo system, whereby insects can synthesise certain saturated fatty acids from other resources, such as carbohydrates and proteins [45]. The total amount of SFAs differed among all insects, with the highest in LM (39.21%) and the lowest in BD (24.64%), values that are higher than the ones reported previously [17]. With regard to MUFAs, high amounts of oleic acid were found in all insects (ranging from 23.30% to 54.26% of the total FA), being the most abundant fatty acid in all insects except for ACD. Palmitoleic acid was only observed in BD (2.90%) and TM (1.32%), as well as 11-eicosenoic acid in GM (2.36%) and nervonic acid in TM (1.08%). Previous studies have also shown higher proportions of 11-eicosenoic acid in triglycerides from males of GM compared to other insects [46]. Due to the high content of oleic acid, a remarkable amount of total MUFAs was found in BD, which reached 58.28% of the total FA, followed by GM with a value of 46.57% of the total FA. A much lower percentage was found in ACD, which was only 25.06%. Linoleic acid was the only PUFA present in all samples, for which the highest value was shown in ACD (36.67%). This may be due to the fact that ACD is able to biosynthesise linoleic acid thanks to the enzyme ∆12 desaturase, which is naturally present in its body [47]. In contrast, α-linolenic acid was present in values above 1% in four out of seven insects, having the highest value by far in LM (10.47%). The latter value is somewhat in line with that observed by Ramos-Bueno et al. [48] (11.5–14.4%), making this species particularly attractive within the food industry as a source for the production of products enriched with n − 3 oils for nutritional and medicinal uses. In line with previous findings, only trace amounts (<1%) of eicosapentaenoic acid (EPA, C20:5n − 3) and docosahexaenoic acid (DHA, C22:6n − 3) were found in ACD, ALD, and LM, and GM, LM, and TM, respectively [17,43,49,50]. The highest values of total PUFAs were found in ACD (39.76%), followed by TM (30.41%) and LM (27.77%), while the minimum was shown in BD (17.08%) and GM (18.64%).

### 3.2. Nutritional Quality of the Lipids

From the fatty acid profile, it is possible to estimate the nutritional quality of the lipid and the possible effects that its intake may have on the health of consumers. For this purpose, different indices can be used, among which PUFA/SFA and n − 6/-n − 3 ratios, IA, IT, and h/H have been considered in this study (Table 1). For a healthy diet, the consumption of lipids with a PUFA/SFA ratio close to 1 is recommended since it has been observed that the consumption of food with values higher than 1 is related to cardiovascular risk reduction [51]. The consumption of lipids with ratios above 3 could promote the formation of tumours, while ratios below 0.33 in the diet could have an atherogenic effect [52]. Accordingly, edible insects can be considered a healthy lipid source, among which ACD and TM provided the best PUFA/SFA ratio among the species studied (1.1 and 0.98, respectively), which are slightly below the ones reported by Otero et al. [53]. Together with that, a high dietary n − 6/n − 3 value may be related to the risk of developing coronary heart disease and cancer, among others; ratios below 4 are advisable [54]. Of the edible insects examined in this work, only LM exhibits a value below the recommendation (1.59), mainly due to its high α-linolenic acid content, as mentioned above. Still, our results were far lower than the ones obtained by Paul et al. [47] for TM, with reported values of approximately 200, which the authors suggested to be due to the low intake of n − 3 fatty acids in the feed. While Cito et al. [55] showed values ranging between 1.98 and 4.86 for GM, which were in line with those obtained by Francardi et al. [56], our sample reached a value of 13.76, even substantially higher than the 7.3 reported by Guil-Guerrero et al. [19]. As stated before, there have been suggestions that these differences may be due to the fact that fatty acid profiles depend on extrinsic factors such as feed [11,47,55]. However, more recent studies have revealed that, regardless of the diet of the insects, the concentration of the main fatty acids remained relatively stable, and changes were essentially limited to n − 3 PUFAs [50]. Furthermore, it has been observed that TM has the ability to self-select nutrients from the feed based on its physiological needs. Thus, a higher availability of particular nutrients in the feed does not necessarily result in an increased level of these nutrients in the insect [57]. It is reasonable to assume that more species may also have that ability, which could suggest that physiological aspects may play a more important role in the lipid profile compared to the feed.

The IA, IT, and h/H index also showed significant differences among species. While lower values of IA and IT are related to a lower risk of developing cardiovascular diseases, higher levels of h/H are considered desirable [58]. In this study, the IA index ranged from 0.26 to 0.58, which is comparable to those reported for meat and fish and lower than those observed for dairy products, while the IT, ranging from 0.57 to 1.08, was higher than those reported for most fish, in the same range as those reported for most meat and lower than those observed for dairy [58]. Likewise, the h/H ranged from 1.86 to 4.20, which is in line with values previously reported in insects [31,32], higher than those reported for dairy products, and in the same range as most meat and fish [58]. BD reached the lowest IA and IT indices (0.26 and 0.57, respectively) with a considerably higher value of h/H (4.20), suggesting the fatty acid profile of this insect could be the healthiest among the insects studied. In contrast, LM exhibited a higher IA (0.58) compared to the other insects and a rather low h/H (1.96). ZM showed the highest IT (1.08), a rather high IA (0.52), and a rather low h/H (2.06). Even so, the IT value obtained for ZM was slightly below that previously reported for 60-day-old larvae (1.19) [59]. On the other hand, the IT value obtained for ALD was somewhat higher than that reported by Mohammad Taghi Gharibzahedi and Altintas [31], but for TM, it was considerably higher than that obtained by Lawal et al. [32]. According to the evidence reported by Mohammad Taghi Gharibzahedi and Altintas [31], these indices may be influenced by the solvent used during lipid extraction, explaining the variation among these different reports. In our study, lipids were extracted with chloroform/methanol, which allows a quantitative extraction of all lipids [60], hence the indices most accurately reflect the lipid composition of the entire insect as a whole. More importantly, as mentioned above, diet may also be a factor to consider in such differences.

### 3.3. Analysis of Lipid Classes

Lipids were separated into lipid classes in order to have a more detailed knowledge of the lipid composition of the selected edible insects. In Figure 1, the amount of neutral lipids, glycolipids, and phospholipids that constitute the total lipid is presented, expressed in g/100 g DM (average values and the corresponding SEM are also provided in ST1). Non-polar or neutral lipids, with values ranging between 46.56% in ACD and 86.15% in GM of total lipid content, were the highest fraction in all insects, as this lipid group serves as an energy deposit in the form of triacylglycerols [16]. The content of polar fractions, glycolipids, and phospholipids reached values up to 10.61 g/100 g DM of glycolipids in ZM and 5.54 g/100 g DM of phospholipids in ACD. Phospholipids play a fundamental role in the structure of cell membranes. This lipid fraction represented 9.9% of the total lipids in TM, a significantly lower percentage than previously reported by Gowda et al. [15] (51%). Nevertheless, it has been shown on insects typically consumed in Nigeria that although the phospholipid fraction was the second most abundant, values remained below 20% [61]. Among the insects used in this study, only BD and LM displayed a phospholipid fraction as the second most abundant (24.64 and 14.79% of total lipid content, respectively). The presence of phospholipids has also been confirmed by Tzompa-Sosa et al. [17] in lipid extracts of ACD, ALD, BD, and TM. They observed that the extraction method affects the composition of the extracts as the phospholipids become water-soluble and remain in the aqueous layer, thus not being isolated when lipid separation in aqueous conditions is applied. Hence, using organic solvents in this study allows us to obtain the phospholipidic fraction. On the other hand, TM, ALD, and ZM, which belong to the family Tenebrionidae, were the insects with the highest percentage of glycolipids with respect to total lipids (42, 38, and 31%, respectively), which could be interesting not only for agrifood uses but also for the cosmetics industries for the development of skin treatment products [62].

It has been suggested that the sensitivity of fatty acids to oxidation may depend on their incorporation in the lipids, i.e., either free or esterified in the form of triacylglycerols or other complex lipids, such as phospholipids and glycolipids [20,21,63]. Similarly, the digestion and absorption of fatty acids may vary depending on the lipid class in which they are incorporated [64]. Since there is limited research on the issue, it was considered valuable to know the specific fatty acid composition of the lipid classes as a basis for further understanding of their oxidative stability and bioavailability. The fatty acid profiles obtained from the neutral lipid, glycolipid, and phospholipid fractions, expressed as the ratio of the specific fatty acid against the sum of all FAMEs in the lipid class, are shown in Table 2, Table 3 and Table 4, respectively. As expected, since it is the major class (from 46.56% in ACD to 86.15% in GM of the total lipids), the fatty acid composition of the neutral fraction is similar to the composition of the total lipids; only small differences can be observed. Likewise, the glycolipids showed slight variations in fatty acid composition compared to the total lipids. However, the phospholipid fraction showed the greatest variation relative to the total lipid fraction. All insects presented a considerably lower content of oleic acid in the phospholipid fraction but a higher percentage of stearic acid compared to the total lipid fraction. Linoleic acid showed higher percentages in the phospholipid fraction in most of the insects, except for ALD and TM. While in ALD similar values were recorded, the phospholipid fraction of TM, in contrast to the rest of the insects studied, was the only one that exhibited strikingly lower values of linoleic acid as well as a higher content of palmitic acid in comparison to the total lipid profile. LM showed a considerably high content of α-linolenic acid, which is an essential fatty acid, in the phospholipid fraction. In addition, the phospholipid fraction was the only fraction in which DHA content was higher than 1%, more specifically in TM, suggesting that the trace levels observed in the total lipid profile were derived from its phospholipid fraction. It is known that the bioavailability of fatty acids depends on many factors, such as their chemical structure and the food matrix, among others, and may even be influenced by the lipid class to which they belong [65]. However, there is still substantial debate on this, and further research, including in vivo studies, is needed to substantiate the role of lipid class in the bioavailability of semi-essential fatty acids such as α-linolenic acid, EPA, and DHA.

As noted above, lipid oxidation is one of the main reasons for chemical food degradation. Unsaturated fatty acids, and especially PUFA, are more susceptible to oxidation than saturated fatty acids due to their lower energy of activation at the initial stage when free radicals are formed [66]. Despite the presence of PUFA in all three fractions, in most of the insects studied, they were the most abundant in the phospholipid fraction (except for ALD and TM). The same pattern has been previously reported in vegetables [20]. According to previous studies [20,67], the phospholipid fraction may be more protective against auto-oxidation than the neutral and glycolipid fractions but is more rapidly oxidised by photo-oxidation and lipoxygenase-catalysed oxidation [21].

### 3.4. Amino Acid Composition and Estimation of Protein Quality

The total amounts and composition of amino acids are shown in Table 5, except for tryptophan, as alkaline hydrolysis was not performed. As previous studies reported, insects were considered rich in glutamic acid and, in general, present great amounts of alanine and aspartic acid but are poor in cysteine [11,13,49]. According to that, in all the samples analysed, cysteine was the AA with the lowest values (0.73–1.46 g/100 g of protein), whilst the sum of glutamic acid and glutamine showed the highest content (11.28–13.74 g/100 g of protein), followed by alanine in ACD, BD, and LM (9.17–12.73 g/100 g of protein), asparagine and aspartic acid in ALD and GM (9.47 and 10.14 g/100 g of protein, respectively), and tyrosine in TM and ZM (8.11 and 9.23 g/100 g of protein, respectively). The total sulphur-containing and total aromatic amino acid content in all insects reached 2.29–3.65 g/100 g of protein for the former and 8.87–14.11 g/100 g of protein for the latter. Among insects, ZM showed the greatest content of aromatic amino acids (tyrosine and phenylalanine), while ACD presented the highest content of sulphur amino acids (methionine and cysteine).

Protein quality relies on both constituent amino acids and protein digestibility [68]. While studying digestibility was beyond the scope of this paper, the amino acid composition of the studied insects was investigated together with common nutritional ratios based on the amino acid profile, which include total EAA content, E/T (%), EAAI, PER, and AAS (Table 5).

The total EAA content ranged between 39.57 g/100g of protein in GM and 49.28 g/100g of protein in ZM. Among them, values for ACD, BD, and LM were comparable to those in the literature [11,13,44,49,69]. In contrast, while Finke [11] reported similar values for ZM and TM (about 48% for both insects), Yi et al. [13] reported lower content (44%) for ZM and, to a lesser extent, in TM (46%). On the other hand, the EAA values in GM and ALD were slightly lower than those previously reported (39.57 and 46.44% versus 43 and 48%, respectively) [11,13,49]. Apart from that, while EAA values were in line with those reported for other insect species, such as *A. mellifera* and *S. gregaria* (43.8 and 39.6 g/100 g of protein, respectively), the reported values of conventional protein sources such as eggs (43.6 g/100 g of protein) or milk (42.6 g/100 g of protein) were lower than those obtained for insects like ZM, TM, and ALD. This highlights their relevance as alternative protein sources [70], together with the fact that the percentage ratios of essential to total amino acids (% E/T) reached values from 39.57% in GM to 49.28% in ZM, all values higher than the 36% that was considered appropriate for an ideal protein [71].

Regarding EAAI, a higher value represents higher protein efficiency and quality [72]. Our data presented values ranging between 2.43 and 2.51. While these values were lower than those reported for other insects, such as P. reticularis (3.3–3.4) [39], all were higher than 1, which suggested that insect amino acid composition is superior to the FAO/WHO [39] standard and thus insect protein can be considered a good-quality protein [39].

In addition, AAS is an index to evaluate the protein and amino acid efficiency that is required for the various population groups, based on the fact that protein synthesis in the body cells does not take place unless the necessary amino acids are provided by the diet. Therefore, the AAS represents the protein quality regarding the EAA proportion present in a food [73]. Based on our analysis, Leu was the limiting amino acid for ALD, DB, and GM, whereas for ACD, LM, and TM, the limiting amino acids were Lys and Met + Cys for ZM. Despite this, the AAS for all insects (111–159%) could reach the FAO/WHO requirements [39] for older children, adolescents, and adults.

Lastly, PER estimates the dietary value of different proteins by calculating it based on the interaction between Leu-Pro and Leu-Tyr (PER-1 and PER-2). According to Sommano et al. [73], PER values lower than 1.5 correspond to low-quality proteins, while those higher than 2 correspond to high-quality proteins. The values obtained for PER-1 ranged between 1.94 and 2.78, while those for PER-2 varied from 1.83 to 2.77. In this regard, noteworthy was the score of LM (2.78 and 2.77 for PER-1 and PER-2, respectively), which was higher than those of other insects, suggesting that LM might have higher digestibility [74]. Even so, the predicted PER-1 and PER-2 values of all insects were above 1.5, even in most cases above 2, which makes insect protein among the high-quality proteins. However, these marks were slightly lower than those previously observed for meat products and much lower than those for fish (2.7–3.2 and 3.1–3.7, respectively) [68].

Nevertheless, amino acid availability varies according to protein source, processing, and interaction with other food constituents, which all affect protein digestibility [75]. Hence, it should be noted that these data only provide an overview of the potential nutritional value of the protein and should be confirmed by further experimental trials in vivo [76], which were not the focus of the present study.

### 3.5. True Protein Content

Based on the amino acid content, the true protein content was calculated (ST1). Our findings showed that the highest protein content was found in BD (61.49 g/100 g DM), followed by ALD (55.74 g/100 g DM), and TM and LM (44.68 g/100 g and 44.23 g/100 g DM, respectively). A lower amount was found in ACD (39.37 g/100 g DM), followed by ZM (34.57 g/100 g DM), and the lowest values were shown in GM (28.69 g/100 g DM). As was expected, these true protein values are, in general, lower than the crude protein contents reported previously by other authors since, in most studies, crude protein calculation was based on the application of Jones’ conversion factor (6.25) to the total nitrogen content [43,44,59], as discussed further below.

### 3.6. Protein Conversion Factor (Kp)

There is some controversy concerning the Kp that should be used to calculate protein content from nitrogen analysis. Previous research, due to insufficient data, often used the default value for animal protein (6.25). However, for insects, the total protein value could be overestimated owing to the non-protein nitrogen derived from other compounds, such as chitin [37]. In order to give a precise and accurate result, the true protein content reported above was used to calculate the specific Kp for each insect (Figure 2). As already pointed out, the calculated Kp showed lower values than the commonly used 6.25, with the exception of ALD (Kp = 6.43). Apart from ALD, GM, TM, and LM showed the highest values (5.25, 5.20, and 5.15, respectively), while the lowest values were found in ACD, BD, and ZM, with comparable Kp values in the range of 4.17 to 4.90. As stated before, it is clear that the use of the common Jones’ factor (6.25) significantly overestimates the protein content in six of the seven insects studied, from 19% in the case of GM up to as much as 50% in the case of ACD. Values for LM were similar to those obtained previously by Boulos et al. [26] (5.33). By contrast, values for ACD were slightly lower than those obtained by Belghit et al. [24] (4.53–4.80) and substantially lower than the 5.09 reported by Ritvanen et al. [27] and the 5.25 shown by Boulos et al. [26]. On the other hand, the values for TM were similar to those obtained previously by Boulos et al. [26] (5.41) but were slightly higher than those obtained by Belghit et al. [24], as well as those obtained by Janssen et al. [25], who observed Kp of 4.64–4.86. Likewise, somewhat larger values were obtained by Belghit et al. [24] and Janssen et al. [25] for ALD, which reached 4.86–5.05. These higher values could be explained by the different methods used for total nitrogen determination. Both studies, Belghit et al. [24] and Janssen et al. [25], analysed total nitrogen by the Dumas method, which has been observed to lead to higher nitrogen levels than when the Kjeldahl method is used [34], whereby a higher total nitrogen results in a lower Kp.

To the best of our knowledge, our study is the first publication in which the Kp of BD, GM, and ZM is calculated, which provides useful input when used as an alternative source of protein. Apart from that, the range of Kp found in all these insects was in line with those reported for other insects, such as *Apis mellifera* pupae (4.9) and *Schistocerca gregaria* (4.5) [25,37,77]. Boulos et al. [26] suggested an average of 5.33 as the factor that was the most accurate indicator of the total protein content of the insects. Conversely, this study showed that even insects of the same family do not display comparable values. Therefore, it would be inadequate to assume a single value for all insects, making further studies required.

## 4. Conclusions

The nutritional quality indices revealed lipids with an overall favourable AI, TI, and h/H, combined with a good PUFA/SFA ratio, making them a healthy novel lipid source. In terms of these indices, BD can overall be considered to have the healthiest fatty acid composition of the species studied. Furthermore, concerning protein, it was confirmed that all the insects studied exceeded the EAA requirements, with LM showing the best PER values. Still, it is important to emphasise that the fatty acid composition also depends on extrinsic factors, such as the feed. In contrast, the amino acid composition generally showed less variation.

As far as the former is concerned, as expected, neutral lipids constituted the largest fraction. The overall low percentage of phospholipids, BD and LM, and, to a lesser extent, ACD, presented remarkably higher values than the other insects. Apart from ALD and TM, higher percentages of linoleic acid were observed in the phospholipid fraction compared to the total lipid fraction. In addition, LM showed a considerably high proportion of α-linolenic acid in the phospholipid fraction. In TM, DHA was quantifiable in the phospholipid fraction. The presence of these semi-essential fatty acids in phospholipids is noteworthy, given that their absorption may be facilitated when they are found in this class of lipids. Similarly, it has been suggested that PUFA may be somewhat protected against auto-oxidation when they are contained in phospholipids, an interesting fact since most insects have a higher PUFA content in phospholipids than in the rest of the lipid classes. On the other hand, regarding the species-specific Kp, it was confirmed that using the Jones’ factor leads to overestimation of the protein content, which could reach even 50%. It was also found that Kp is independent of the insect family, resulting in a wide range from 4.17 in ACD to 6.43 in ALD. Additionally, data regarding the Kp of BD, GM, and ZM, which corresponded to 4.68, 5.25, and 4.9, respectively, were provided for the first time.

Overall, these data should be complemented by in vivo studies to investigate semi-essential fatty acid bioavailability and protein digestibility.

## Figures and Tables

**Figure 1 foods-12-04090-f001:**
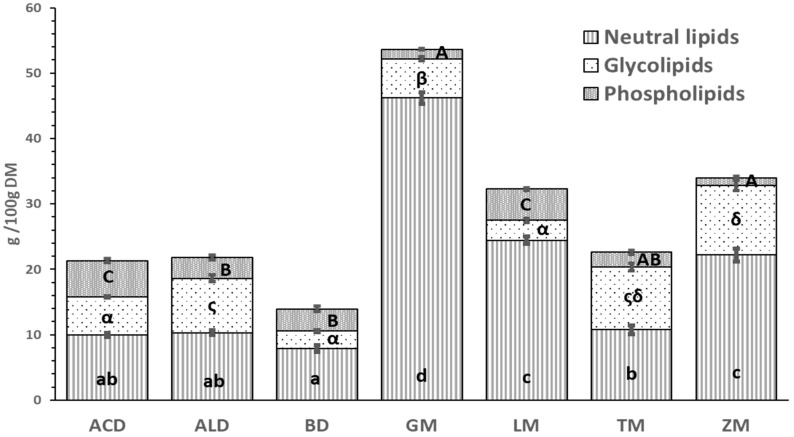
The neutral lipid, glycolipid, and phospholipid fractions of seven edible insects expressed as g/100 dry matter (DM). Mean values corresponding to the same lipid fraction not followed by a common letter (a–d for phospholipid fraction; α, ς, and δ for glycolipid fraction; and A–C for neutral lipids) differ significantly (*p* < 0.05).

**Figure 2 foods-12-04090-f002:**
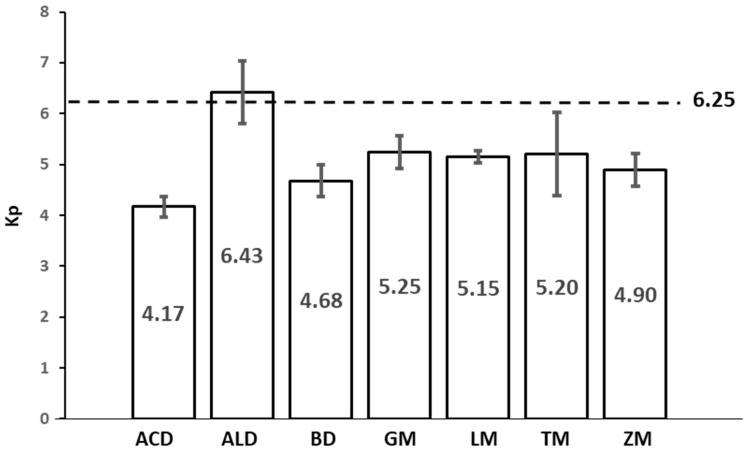
Comparison of the Jone’s conversion factor (6.25) and the means of each insect *Kp* factor (ACD: *A. domesticus;* ALD: *A. diaperinus;* BD: *B. dubia*; GM: *G. mellonella;* LM: *L. migratoria;* TM: *T. molitor;* ZM: *Z. morio)*. The *Kp* factors were calculated based on the true protein content, calculated as the sum of the anhydrous amino acid residues.

**Table 1 foods-12-04090-t001:** The fatty acid profile of the total lipid content of edible insects expressed as a percentage (mean ± SEM, *n* = 10).

FA	ACD	ALD	BD	GM	LM	TM	ZM
C8:0	-	-	-	-	-	-	tr
C9:0	tr	tr	tr	tr	tr	tr	tr
C10:0	tr	tr	tr	tr	tr	tr	tr
C11:1	-	-	-	-	-	-	tr
C12:0	tr	tr	tr	tr	tr	tr	tr
C13:0	-	-	-	-	-	tr	-
C13:1	-	-	tr	-	tr	-	tr
C14:0	tr	tr	tr	tr	1.67 ± 0.01 ^a^	3.57 ± 0.01 ^b^	tr
C14:1	-	-	tr	-	tr	tr	tr
C15:0	tr	tr	tr	tr	tr	tr	tr
C16:0	24.26 ± 0.03 ^c^	24.29 ± 0.16 ^c^	15.95 ± 0.03 ^a^	32.86 ± 0.56 ^e^	28.44 ± 0.17 ^d^	18.62 ± 0.33 ^b^	28.24 ± 0.10 ^d^
C16:1	tr	tr	2.90 ± 0.05 ^b^	tr	tr	1.32 ± 0.00 ^a^	tr
C17:0	tr	tr	tr	tr	tr	tr	tr
C16:3/C17:1	-	tr	tr	tr	tr	tr	tr
C18:0	9.19 ± 0.06 ^g^	8.70 ± 0.12 ^f^	6.02 ± 0.09 ^c^	1.52 ± 0.01 ^a^	8.27 ± 0.14 ^e^	5.33 ± 0.24 ^b^	7.34 ± 0.09 ^d^
C18:1	23.30 ± 0.11 ^a^	37.16 ± 0.09 ^d^	54.26 ± 0.10 ^f^	42.90 ± 0.54 ^e^	31.60 ± 0.17 ^b^	35.63 ± 0.07 ^c^	36.11 ± 0.05 ^c^
C18:2	36.67 ± 0.14 ^e^	22.57 ± 0.10 ^c^	15.66 ± 0.13 ^a^	17.32 ± 0.25 ^b^	16.96 ± 0.08 ^b^	28.23 ± 0.36 ^d^	22.36 ± 0.27 ^c^
C19:0	-	tr	tr	-	-	tr	-
C18:3n − 3	2.55 ± 0.01 ^b^	tr	tr	1.23 ± 0.01 ^a^	10.47 ± 0.11 ^c^	1.18 ± 0.31 ^a^	1.23 ± 0.03 ^a^
C20:0	tr	tr	tr	tr	tr	tr	tr
C20:1	tr	tr	tr	2.36 ± 1.38	tr	tr	-
C20:2	tr	tr	tr	-	-	tr	-
C20:3n − 6	-	tr	-	-	-	-	-
C20:4n − 6	-	tr	tr	-	-	-	-
C20:3n − 3	-	tr	tr	-	-	tr	-
C20:5n − 3	tr	tr	-	-	tr	-	-
C22:0	tr	tr	tr	tr	tr	tr	-
C22:1	tr	tr	tr	tr	tr	tr	-
C22:2	-	-	tr	-	-	-	-
C22:5n − 6	tr	tr	tr	-	tr	tr	-
C22:4	tr	-	-	-	-	-	-
C22:5n − 3	-	tr	tr	tr	tr	-	-
C24:0/C22:6n − 3	-	-	-	tr	tr	tr	-
C24:1	tr	tr	tr	tr	tr	1.08 ± 0.08	tr
SFA	35.18 ± 0.09 ^c^	36.41 ± 0.11 ^d^	24.64 ± 0.13 ^a^	34.79 ± 0.58 ^c^	39.21 ± 0.01 ^f^	31.17 ± 0.34 ^b^	38.11 ± 0.31 ^e^
MUFA	25.06 ± 0.05 ^a^	39.21 ± 0.06 ^d^	58.28 ± 0.06 ^f^	46.57 ± 0.85 ^e^	33.02 ± 0.21 ^b^	38.42 ± 0.03 ^cd^	37.95 ± 0.13 ^c^
PUFA	39.76 ± 0.14 ^f^	24.38 ± 0.16 ^c^	17.08 ± 0.16 ^a^	18.64 ± 0.26 ^b^	27.77 ± 0.21 ^d^	30.41 ± 0.32 ^e^	23.94 ± 0.27 ^c^
PUFA/SFA	1.13 ± 0.01 ^f^	0.67 ± 0.01 ^c^	0.69 ± 0.01 ^cd^	0.54 ± 0.00 ^a^	0.71 ± 0.01 ^d^	0.98 ± 0.02 ^e^	0.63 ± 0.01 ^b^
n − 3	2.75 ± 0.01 ^d^	1.15 ± 0.05 ^b^	0.89 ± 0.00 ^a^	1.26 ± 0.02 ^bc^	10.67 ± 0.15 ^e^	1.49 ± 0.16 ^c^	1.23 ± 0.03 ^b^
n − 6	37.01 ± 0.13 ^f^	23.09 ± 0.10 ^d^	16.05 ± 0.15 ^a^	17.32 ± 0.25 ^b^	16.97 ± 0.09 ^b^	28.77 ± 0.15 ^e^	22.36 ± 0.27 ^c^
n − 6/n − 3	13.47 ± 0.02 ^b^	20.16 ± 0.76 ^d^	18.04 ± 0.18 ^c^	13.76 ± 0.12 ^b^	1.59 ± 0.02 ^a^	19.43 ± 1.92 ^cd^	18.24 ± 0.53 ^cd^
IA	0.41 ± 0.00 ^b^	0.44 ± 0.00 ^c^	0.26 ± 0.00 ^a^	0.51 ± 0.01 ^e^	0.58 ± 0.00 ^f^	0.48 ± 0.01 ^d^	0.52 ± 0.00 ^e^
IT	0.86 ± 0.00 ^d^	0.98 ± 0.01 ^e^	0.57 ± 0.00 ^a^	0.96 ± 0.02 ^e^	0.67 ± 0.00 ^b^	0.72 ± 0.03 ^c^	1.08 ± 0.01 ^f^
h/H	2.54 ± 0.01 ^e^	2.44 ± 0.02 ^d^	4.20 ± 0.01 ^g^	1.86 ± 0.01 ^a^	1.96 ± 0.01 ^b^	2.94 ± 0.05 ^f^	2.06 ± 0.01 ^c^

^a–g^ Mean values in the same row (corresponding to the same fatty acid) not followed by a common letter differ significantly (*p* < 0.05). tr corresponds to a percentage below 1%. ACD: *A. domesticus*; ALD: *A. diaperinus*; BD: *B. dubia*; GM: *G. mellonella*; LM: *L. migratoria*; TM: *T. molitor*; ZM: *Z. morio*; FA: Fatty acid; SFA: Saturated fatty acids; MUFA: Monounsaturated fatty acids; PUFA: Polyunsaturated fatty acids; IA: Index of Atherogenicity; IT: Index of Thrombogenicity; h/H: Hypocholesterolemic/Hypercholesterolemic ratio.

**Table 2 foods-12-04090-t002:** The fatty acid profile from the neutral lipid fraction of edible insects expressed as a percentage (mean ± SEM, *n* = 10).

FA	ACD	ALD	BD	GM	LM	TM	ZM
C8:0	-	-	-	-	-	-	tr
C9:0	tr	tr	tr	tr	tr	tr	tr
C10:0	tr	tr	tr	tr	tr	tr	tr
C11:1	-	-	-	-	-	-	tr
C12:0	tr	-	tr	-	tr	tr	tr
C13:0	-	-	-	-	-	tr	-
C13:1	-	-	-	-	-	-	tr
C14:0	tr	tr	tr	tr	1.88 ± 0.02 ^a^	2.68 ± 0.07 ^b^	tr
C14:1	-	-	tr	-	tr	-	-
C15:0	tr	tr	tr	tr	tr	tr	tr
C16:0	25.91 ± 0.09 ^d^	22.66 ± 0.25 ^c^	17.99 ± 0.04 ^b^	33.93 ± 0.59	30.61 ± 0.23 ^f^	16.13 ± 0.02 ^a^	27.95 ± 0.08 ^e^
C16:1	tr	tr	3.36 ± 0.04 ^b^	tr	tr	1.28 ± 0.01 ^a^	tr
C17:0	tr	tr	tr	tr	tr	tr	tr
C16:3/C17:1	-	tr	tr	tr	tr	tr	tr
C18:0	7.35 ± 0.00 ^e^	7.69 ± 0.06 ^f^	4.34 ± 0.05 ^c^	1.01 ± 0.01 ^a^	6.87 ± 0.19 ^d^	3.74 ± 0.12 ^b^	6.94 ± 0.03 ^d^
C18:1	29.28 ± 0.17 ^a^	42.48 ± 0.26 ^e^	58.11 ± 0.14	43.85 ± 0.61 ^f^	34.18 ± 0.25 ^b^	41.42 ± 0.24 ^d^	37.87 ± 0.07 ^c^
C18:2	30.59 ± 0.08 ^e^	21.68 ± 0.22 ^d^	12.03 ± 0.06 ^a^	16.01 ± 0.27 ^c^	15.08 ± 0.10 ^b^	31.46 ± 0.07 ^f^	21.69 ± 0.04 ^d^
C19:0	-	-	-	-	-	-	-
C18:3n − 3	2.54 ± 0.01 ^c^	tr	tr	1.00 ± 0.02 ^ab^	9.21 ± 0.12 ^d^	tr	1.19 ± 0.04 ^b^
C20:0	-	tr	tr	-	-	tr	-
C20:1	tr	tr	-	2.53 ± 1.52	-	-	-
C20:2	-	-	-	-	-	-	-
C20:3n − 6	-	tr	-	-	-	-	-
C20:4n − 6	-	-	-	-	-	-	-
C20:3n − 3	-	-	-	-	-	-	-
C20:5n − 3	-	tr	-	-	-	-	-
C22:0	-	-	-	-	-	-	-
C22:1	-	-	-	-	-	-	-
C22:2	-	-	-	-	-	-	-
C22:5n − 6	tr	tr	tr	-	-	tr	-
C22:4	tr	-	-	-	-	-	-
C22:5n − 3	-	tr	-	-	-	-	-
C24:0/C22:6n − 3	-	-	-	-	-	-	-
C24:1	1.43 ± 0.04	tr	tr	tr	tr	tr	tr
SFA	34.58 ± 0.03 ^c^	32.08 ± 0.11 ^b^	24.17 ± 0.11 ^a^	35.26 ± 0.61 ^c^	39.92 ± 0.09 ^e^	23.68 ± 0.11 ^a^	36.91 ± 0.24 ^d^
MUFA	31.85 ± 0.10 ^a^	44.27 ± 0.25 ^d^	62.48 ± 0.11 ^f^	47.67 ± 0.90 ^e^	35.68 ± 0.29 ^b^	43.58 ± 0.24 ^d^	39.87 ± 0.20 ^c^
PUFA	33.57 ± 0.07 ^f^	23.64 ± 0.19 ^c^	13.35 ± 0.10 ^a^	17.07 ± 0.29 ^b^	24.40 ± 0.25 ^d^	32.74 ± 0.13 ^e^	23.22 ± 0.04 ^c^

^a–f^ Mean values in the same row (corresponding to the same fatty acid) not followed by a common letter differ significantly (*p* < 0.05). tr corresponds to a percentage below 1%. ACD: *A. domesticus;* ALD: *A. diaperinus;* BD: *B. dubia*; GM: *G. mellonella;* LM: *L. migratoria;* TM: *T. molitor;* ZM: *Z. morio;* FA: Fatty acid; SFA: Saturated fatty acids; MUFA: Monounsaturated fatty acids; PUFA: Polyunsaturated fatty acids.

**Table 3 foods-12-04090-t003:** The fatty acid profile from the glycolipid fraction of edible insects expressed as a percentage (mean ± SEM, *n* = 10).

FA	ACD	ALD	BD	GM	LM	TM	ZM
C8:0	-	-	-	-	-	-	tr
C9:0	tr	-	tr	-	tr	tr	tr
C10:0	-	tr	-	tr	tr	-	tr
C11:1	-	-	-	-	-	-	-
C12:0	tr	tr	tr	tr	tr	tr	-
C13:0	-	-	-	-	-	tr	-
C13:1	-	-	tr	-	tr	-	-
C14:0	tr	1.13 ± 0.00 ^a^	1.18 ± 0.01 ^a^	tr	2.11 ± 0.01 ^c^	4.82 ± 0.05 ^d^	1.29 ± 0.03 ^b^
C14:1	-	-	tr	-	-	tr	tr
C15:0	tr	tr	1.01 ± 0.04	tr	1.20 ± 0.04	tr	tr
C16:0	28.88 ± 0.13 ^d^	26.17 ± 0.03 ^c^	16.94 ± 0.05 ^a^	25.92 ± 0.31 ^c^	29.45 ± 0.10 ^e^	19.64 ± 0.05 ^b^	28.65 ± 0.29 ^d^
C16:1	tr	1.20 ± 0.00 ^b^	3.89 ± 0.02 ^d^	tr	tr	1.67 ± 0.00 ^c^	1.06 ± 0.06 ^a^
C17:0	tr	tr	tr	tr	-	tr	tr
C16:3/C17:1	-	tr	tr	tr	tr	tr	tr
C18:0	7.08 ± 0.18 ^d^	6.47 ± 0.02 ^c^	5.43 ± 0.05 ^b^	3.60 ± 0.12 ^a^	8.08 ± 0.03 ^e^	3.45 ± 0.01 ^a^	6.53 ± 0.23 ^c^
C18:1	23.92 ± 0.22 ^a^	37.96 ± 0.12 ^e^	50.87 ± 0.14 ^g^	39.21 ± 0.20 ^f^	30.48 ± 0.09 ^b^	36.41 ± 0.09 ^d^	34.72 ± 0.36 ^c^
C18:2	33.45 ± 0.24 ^f^	23.57 ± 0.03 ^c^	16.96 ± 0.06 ^b^	25.49 ± 0.23 ^d^	16.08 ± 0.10 ^a^	30.39 ± 0.03 ^e^	23.18 ± 0.21 ^c^
C19:0	-	-	-	-	-	-	-
C18:3n − 3	3.31 ± 0.02 ^d^	tr	1.01 ± 0.02 ^a^	2.42 ± 0.03 ^c^	9.59 ± 0.07 ^e^	tr	1.40 ± 0.03 ^b^
C20:0	tr	tr	tr	-	tr	-	-
C20:1	-	tr	-	1.39 ± 0.62	tr	-	-
C20:2	-	-	-	-	-	-	-
C20:3n − 6	-	-	-	-	-	-	-
C20:4n − 6	-	tr	-	-	-	-	-
C20:3n − 3	-	-	-	-	-	-	-
C20:5n − 3	tr	tr	-	-	tr	-	-
C22:0	-	tr	-	-	tr	-	-
C22:1	-	-	-	-	tr	-	-
C22:2	-	-	tr	-	-	-	-
C22:5n − 6	-	-	-	-	-	-	-
C22:4	-	-	-	-	-	-	-
C22:5n − 3	-	-	tr	-	tr	-	-
C24:0/C22:6n − 3	-	-	-	-	-	-	-
C24:1	tr	tr	1.00 ± 0.11	tr	tr	tr	tr
SFA	37.93 ± 0.05 ^e^	34.98 ± 0.15 ^d^	25.50 ± 0.02 ^a^	30.09 ± 0.23 ^c^	41.75 ± 0.06 ^g^	29.60 ± 0.12 ^b^	38.71 ± 0.08 ^f^
MUFA	25.15 ± 0.22 ^a^	39.88 ± 0.11 ^e^	55.91 ± 0.09 ^g^	41.95 ± 0.45 ^f^	32.23 ± 0.11 ^b^	38.91 ± 0.12 ^d^	36.27 ± 0.31 ^c^
PUFA	36.92 ± 0.27 ^f^	25.15 ± 0.05 ^b^	18.59 ± 0.11 ^a^	27.96 ± 0.23 ^d^	26.02 ± 0.17 ^c^	31.50 ± 0.01 ^e^	25.02 ± 0.23 ^b^

^a–g^ Mean values in the same row (corresponding to the same fatty acid) not followed by a common letter differ significantly (*p* < 0.05). tr corresponds to a percentage below 1%. ACD: *A. domesticus;* ALD: *A. diaperinus;* BD: *B. dubia*; GM: *G. mellonella;* LM: *L. migratoria;* TM: *T. molitor;* ZM: *Z. morio;* FA: Fatty acid; SFA: Saturated fatty acids; MUFA: Monounsaturated fatty acids; PUFA: Polyunsaturated fatty acids.

**Table 4 foods-12-04090-t004:** The fatty acid profile from the phospholipid fraction of edible insects expressed as a percentage (mean ± SEM, *n* = 10).

FA	ACD	ALD	BD	GM	LM	TM	ZM
C8:0	-	-	-	-	-	-	-
C9:0	tr	-	tr	-	-	5.00 ± 3.92	-
C10:0	tr	-	tr	-	tr	1.66 ± 0.90	-
C11:1	-	-	-	-	-	-	-
C12:0	-	tr	-	tr	-	1.02 ± 0.34	-
C13:0	-	-	-	-	-	-	-
C13:1	-	-	-	-	-	-	-
C14:0	tr	tr	tr	tr	tr	2.50 ± 0.37	-
C14:1	-	-	-	-	-	-	-
C15:0	tr	1.78 ± 0.07 ^a^	tr	tr	tr	3.78 ± 0.22 ^b^	tr
C16:0	16.27 ± 0.18 ^a^	24.63 ± 1.28 ^b^	10.53 ± 0.02 ^a^	27.31 ± 2.02 ^b^	16.66 ± 0.21 ^a^	26.09 ± 2.62 ^b^	29.98 ± 5.02 ^b^
C16:1	tr	tr	1.07 ± 0.15	tr	tr	-	-
C17:0	tr	tr	tr	tr	-	1.17 ± 0.15	tr
C16:3/C17:1	-	-	-	-	tr	-	-
C18:0	14.86 ± 0.17 ^b^	17.85 ± 0.96 ^c^	10.31 ± 0.27 ^a^	9.10 ± 0.71 ^a^	15.59 ± 0.04 ^b^	21.28 ± 1.23 ^d^	21.91 ± 0.54 ^d^
C18:1	11.91 ± 0.06 ^b^	17.74 ± 0.71 ^cd^	48.03 ± 0.17 ^f^	27.86 ± 1.41 ^e^	19.15 ± 0.10 ^d^	4.13 ± 0.26 ^a^	15.83 ± 2.02 ^c^
C18:2	50.86 ± 0.47 ^c^	22.84 ± 0.57 ^b^	22.99 ± 0.35 ^b^	25.59 ± 1.19 ^b^	27.13 ± 0.04 ^b^	4.77 ± 4.35 ^a^	27.40 ± 5.28 ^b^
C19:0	-	tr	tr	-	-	tr	-
C18:3n − 3	1.73 ± 0.04 ^a^	tr	tr	3.81 ± 0.28 ^a^	17.47 ± 0.10 ^b^	2.31 ± 3.69 ^a^	tr
C20:0	tr	4.53 ± 0.23 ^b^	1.35 ± 0.03 ^a^	tr	tr	7.99 ± 0.61 ^c^	1.67 ± 1.53 ^a^
C20:1	-	tr	tr	tr	-	tr	-
C20:2	tr	tr	tr	-	-	2.93 ± 4.54	-
C20:3n − 6	-	-	-	-	-	-	-
C20:4n − 6	-	tr	tr	-	-	-	-
C20:3n − 3	-	tr	tr	-	-	tr	-
C20:5n − 3	tr	tr	-	-	-	-	-
C22:0	tr	3.47 ± 2.82	tr	tr	tr	4.48 ± 0.37	-
C22:1	tr	tr	tr	tr	tr	2.77 ± 0.36	-
C22:2	-	-	-	-	-	-	-
C22:5n − 6	-	-	-	-	tr	-	-
C22:4	-	-	-	-	-	-	-
C22:5n − 3	-	-	-	tr	tr	-	-
C24:0/C22:6n − 3	-	-	-	tr	tr	2.97 ± 0.83	-
C24:1	tr	1.74 ± 0.32 ^ab^	1.03 ± 0.01 ^a^	tr	tr	3.23 ± 0.53 ^b^	1.22 ± 1.11 ^a^
SFA	33.34 ± 0.45 ^b^	54.26 ± 0.98 ^c^	25.04 ± 0.28 ^a^	39.28 ± 2.88 ^b^	33.92 ± 0.44 ^b^	75.76 ± 2.94 ^d^	55.20 ± 5.24 ^c^
MUFA	12.96 ± 0.05 ^a^	20.93 ± 0.47 ^b^	50.49 ± 0.07 ^d^	30.35 ± 1.88 ^c^	19.94 ± 0.08 ^b^	10.91 ± 0.67 ^a^	17.05 ± 2.83 ^b^
PUFA	53.70 ± 0.50 ^d^	24.81 ± 0.52 ^b^	24.47 ± 0.35 ^b^	30.37 ± 1.00 ^b^	46.14 ± 0.44 ^c^	13.33 ± 3.06 ^a^	27.76 ± 5.05 ^b^

^a–f^ Mean values in the same row (corresponding to the same fatty acid) not followed by a common letter differ significantly (*p* < 0.05). tr corresponds to a percentage below 1%. ACD: *A. domesticus;* ALD: *A. diaperinus;* BD: *B. dubia*; GM: *G. mellonella;* LM: *L. migratoria;* TM: *T. molitor;* ZM: *Z. morio;* FA: Fatty acid; SFA: Saturated fatty acids; MUFA: Monounsaturated fatty acids; PUFA: Polyunsaturated fatty acids.

**Table 5 foods-12-04090-t005:** Amino acid composition (g/100 g of protein) and nutritional parameters of seven edible insects.

Amino Acid	ACD	ALD	BD	GM	LM	TM	ZM	FAO/WHO (2007) Standard
Met	2.20 ± 0.13 ^b^	1.71 ± 0.08 ^a^	1.57 ± 0.06 ^a^	1.64 ± 0.25 ^a^	1.42 ± 0.21 ^a^	1.75 ± 0.27 ^ab^	1.64 ± 0.03 ^a^	
Cyss + cys	1.46 ± 0.25 ^b^	0.84 ± 0.04 ^a^	1.21 ± 0.08 ^ab^	0.73 ± 0.11 ^a^	0.87 ± 0.19 ^a^	1.15 ± 0.10 ^ab^	0.90 ± 0.36 ^a^	
Total sulphur AA	3.65 ± 0.38 ^b^	2.56 ± 0.11 ^a^	2.78 ± 0.03 ^ab^	2.37 ± 0.36 ^a^	2.29 ± 0.39 ^a^	2.90 ± 0.37 ^ab^	2.55 ± 0.35 ^a^	1.6
Tyr	6.26 ± 0.57 ^ab^	6.16 ± 0.38 ^ab^	7.28 ± 0.78 ^bc^	6.47 ± 0.66 ^ab^	5.64 ± 0.28 ^a^	8.11 ± 0.48 ^cd^	9.23 ± 0.35 ^d^	
Phe	4.07 ± 0.13 ^abcd^	4.22 ± 0.29 ^bcd^	3.69 ± 0.09 ^abc^	3.36 ± 0.61 ^ab^	3.23 ± 0.09 ^a^	4.39 ± 0.50 ^cd^	4.88 ± 0.10 ^d^	
Total aromatic AA	10.33 ± 0.70 ^ab^	10.38 ± 0.67 ^ab^	10.97 ± 0.77 ^ab^	9.83 ± 1.25 ^a^	8.87 ± 0.20 ^a^	12.49 ± 0.98 ^bc^	14.11 ± 0.45 ^c^	3.8
Ile	4.46 ± 0.07 ^b^	4.77 ± 0.03 ^bc^	3.95 ± 0.19 ^a^	3.97 ± 0.34 ^a^	4.66 ± 0.10 ^bc^	4.94 ± 0.06 ^c^	4.91 ± 0.06 ^c^	3
Leu	7.79 ± 0.02 ^cd^	7.36 ± 0.06 ^bc^	6.99 ± 0.16 ^ab^	6.55 ± 0.57 ^a^	8.44 ± 0.03 ^d^	7.88 ± 0.09 ^cd^	7.56 ± 0.13 ^bc^	5.9
Lys	5.20 ± 0.66 ^a^	7.35 ± 0.42 ^b^	6.03 ± 0.65 ^ab^	5.67 ± 0.89 ^a^	5.01 ± 0.31 ^a^	5.20 ± 0.44 ^a^	5.09 ± 0.22 ^a^	4.5
Thr	4.07 ± 0.14 ^ab^	4.42 ± 0.01 ^b^	4.20 ± 0.11 ^ab^	3.70 ± 0.53 ^a^	3.82 ± 0.10 ^ab^	4.28 ± 0.04 ^ab^	4.27 ± 0.04 ^ab^	2.3
Val	6.59 ± 0.24 ^b^	6.40 ± 0.01 ^b^	7.01 ± 0.37 ^b^	5.44 ± 0.52 ^a^	7.10 ± 0.33 ^b^	7.13 ± 0.13 ^b^	6.99 ± 0.05 ^b^	3.9
His	2.92 ± 0.20 ^b^	3.21 ± 0.19 ^bc^	3.14 ± 0.30 ^bc^	2.04 ± 0.40 ^a^	2.87 ± 0.27 ^b^	3.55 ± 0.19 ^bc^	3.80 ± 0.10 ^c^	1.5
Total EAA	45.01 ± 0.02 ^bc^	46.44 ± 0.61 ^bc^	45.07 ± 0.34 ^bc^	39.57 ± 4.83 ^a^	43.06 ± 0.54 ^ab^	48.37 ± 1.24 ^bc^	49.28 ± 0.34 ^c^	
Ser	5.36 ± 0.27 ^ab^	4.66 ± 0.03 ^ab^	4.66 ± 0.09 ^ab^	10.09 ± 5.42 ^b^	4.08 ± 0.10 ^a^	5.05 ± 0.06 ^ab^	4.84 ± 0.04 ^ab^	
Arg	7.78 ± 0.15 ^c^	5.91 ± 0.22 ^ab^	5.71 ± 0.19 ^ab^	5.01 ± 0.74 ^a^	6.20 ± 0.17 ^b^	5.94 ± 0.19 ^b^	5.70 ± 0.11 ^ab^	
Gly	6.17 ± 0.16 ^abc^	5.02 ± 0.15 ^a^	6.97 ± 0.51 ^bc^	6.36 ± 1.54 ^abc^	7.34 ± 0.38 ^c^	5.96 ± 0.03 ^abc^	5.53 ± 0.09 ^ab^	
Asx	8.28 ± 0.28 ^ab^	9.47 ± 0.41 ^c^	9.31 ± 0.68 ^bc^	10.14 ± 0.36 ^c^	7.21 ± 0.31 ^a^	7.88 ± 0.36 ^a^	8.16 ± 0.21 ^a^	
Glx	11.72 ± 0.89	13.43 ± 0.44	12.05 ± 1.05	13.74 ± 1.69	11.28 ± 0.55	11.49 ± 0.43	13.24 ± 0.33	
Ala	9.17 ± 0.46 ^bc^	7.95 ± 0.25 ^ab^	9.68 ± 0.88 ^c^	7.32 ± 0.79 ^a^	12.73 ± 0.31 ^d^	7.65 ± 0.47 ^a^	7.25 ± 0.05 ^a^	
Pro	6.52 ± 0.38 ^ab^	7.13 ± 0.09 ^bc^	6.56 ± 0.28 ^ab^	7.77 ± 0.83 ^c^	8.09 ± 0.31 ^c^	7.66 ± 0.11 ^c^	6.00 ± 0.06 ^a^	
Total NEAA	54.99 ± 0.02 ^ab^	53.56 ± 0.61 ^ab^	54.93 ± 0.34 ^ab^	60.43 ± 4.83 ^c^	56.94 ± 0.54 ^bc^	51.63 ± 1.24 ^ab^	50.72 ± 0.34 ^a^	
E/T (%)	45.01 ± 0.02 ^bc^	46.44 ± 0.61 ^bc^	45.07 ± 0.34 ^bc^	39.57 ± 4.83 ^a^	43.06 ± 0.54 ^ab^	48.37 ± 1.24 ^bc^	49.28 ± 0.34 ^c^	
EAAI	2.48 ± 0.00 ^bc^	2.48 ± 0.01 ^bc^	2.48 ± 0.00 ^bc^	2.43 ± 0.04 ^a^	2.46 ± 0.01 ^ab^	2.50 ± 0.01 ^bc^	2.51 ± 0.00 ^c^	
AAS (%)	132.09 ± 0.41	124.75 ± 0.96	118.54 ± 2.71	111.02 ± 9.71	111.30 ± 6.82	115.49 ± 9.85	159.28 ± 21.71	
Limiting AA	Lys	Leu	Leu	Leu	Lys	Lys	Met + Cys	
PER−1	2.56 ± 0.02 ^cd^	2.34 ± 0.02 ^bc^	2.20 ± 0.09 ^ab^	1.94 ± 0.22 ^a^	2.78 ± 0.03 ^d^	2.55 ± 0.04 ^cd^	2.48 ± 0.06 ^c^	
PER−2	2.41 ± 0.06 ^d^	2.23 ± 0.03 ^bcd^	1.94 ± 0.15 ^ab^	1.83 ± 0.20 ^a^	2.77 ± 0.04 ^e^	2.26 ± 0.04 ^cd^	1.99 ± 0.09 ^abc^	

^a–d^ Mean values in the same row (corresponding to the same volatile compound) not followed by a common letter differ significantly (*p* < 0.001). ACD: *A. domesticus;* ALD: *A. diaperinus*; BD: *B. dubia*; GM: *G. mellonella;* LM: *L. migratoria*; TM: *T. molitor;* ZM: *Z. morio;* His: Histidine; Thr: Threonine; Cyss + cys: Cysteine; Lys: Lysine; Tyr: Tyrosine; Met: Methionine; Val: Valine; Ile: Isoleucine; Leu: Leucine; Phe: Phenilalanine; Ser: Serine; Arg: Arginine; Gly: Glycine; Asx: Sum of asparagine and aspartic acid; Glx: Sum of glutamine and glutamic acid; Ala: Alanine; Pro: Proline; AA: Amino acid; EAA: Essential aminoacids; NEAA: Non-essential amino acids; E/T: Essential to total amino acid ratio; EAAI: Essential amino acid index; AAS: Amino acid score; PER: Predicted protein efficiency ratio.

## Data Availability

Data is contained within the article or Appendix A.

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
