# Peer review of "Fatty Acid and Amino Acid Profiles of Seven Edible Insects: Focus on Lipid Class Composition and Protein Conversion Factors"

_foods, 2023, doi:10.3390/foods12224090_

Round 1
Reviewer 1 Report
Comments and Suggestions for Authors
In my opinion, the manuscript entitled Fatty acid and amino acid profile of 7 edible insects: Focus on lipid class composition and protein conversion factors by Perez-Santaescolastica et al., is a good one, having a high interest for the readers.
The introduction provides enough information regarding the current state of the art, the materials and methods are enough and detailed descripted and the results and discussion are relevant and compared with other studies.
I have only some comments and suggestions:
1. Line 102. There is a red comma; please put it on black color.
2. Lines 242, 247 – please put the scientific denomination of the insects in italic such as A. domesticus and so on.
3. Line 152, please put in italic form the scientific domination of Apis mellifera and gregaria.
4. In conclusion, please only highlight the main findings of the present study, without the introduction part. The conclusion must be short, emphasizing the main results and without the explanation part. Please consider to realize other conclusion and the explanations remove them to the results and discussion part.
Comments on the Quality of English LanguageI believe that there is only minor English errors.
Author Response
Responses to Reviewers
First and foremost, the authors would like to thank the reviewers for taking the time to review the manuscript and make recommendations for improving its quality. Based on their input, we substantially revised the manuscript. Below, the changes made in response to the reviewer comments are presented.
In my opinion, the manuscript entitled Fatty acid and amino acid profile of 7 edible insects: Focus on lipid class composition and protein conversion factors by Perez-Santaescolastica et al., is a good one, having a high interest for the readers.
The introduction provides enough information regarding the current state of the art, the materials and methods are enough and detailed descripted and the results and discussion are relevant and compared with other studies.
Thanks. We highly appreciate your comments.
I have only some comments and suggestions:
- Line 102. There is a red comma; please put it on black color.
Thank you for the remark, the mistake was corrected.
- Lines 242, 247 – please put the scientific denomination of the insects in italic such as domesticus and so on.
Thank you for the remark, the mistake was corrected, see lines 245-246.
- Line 152, please put in italic form the scientific domination of Apis mellifera and gregaria.
Thank you for the remark, the mistake was corrected, see line 513. - In conclusion, please only highlight the main findings of the present study, without the introduction part. The conclusion must be short, emphasizing the main results and without the explanation part. Please consider to realize other conclusion and the explanations remove them to the results and discussion part.
The conclusions was shorted following the recommendation. The introductions part was removed.
Reviewer 2 Report
Comments and Suggestions for Authors
The article with the title "Fatty acid and amino acid profile of 7 edible insects: Focus on lipid class composition and protein conversion factors" has 24 pages with 76 actual references, and 2 Figures and 5 Tables.
Almost half of the resources (30) are used in the Introduction chapter.
I have more comments about the formal editing and writing style of the article, the experiment is sufficiently described from the point of killing of the insect larvae to its analyses.
Abstract
To make a statement about the results, it is better to state the evidence, i.e. whether a significant difference was found or not.
Words from the title of the article are not usually included in the keywords, due to duplication in the search. I would like to mention the names of some representatives of the insects used in the experiment as a hint for improvement.
There are 30 references in the introductory chapter. This is not harmful if their use is not rather declarative, see lines 53 to 55, where 10 references are used without a significant purpose.
In my opinion, the material and methods chapter is more suitable for using at least the same number of citations in the text, as a reference to the fact that the analyzes were used in this case in a comparable way, as is customary.
The results and discussions presented in the next chapter are clear, I have no comments here.
Author Response
Responses to Reviewers
First and foremost, the authors would like to thank the reviewers for taking the time to review the manuscript and make recommendations for improving its quality. Based on their input, we substantially revised the manuscript. Below, the changes made in response to the reviewer comments are presented.
The article with the title "Fatty acid and amino acid profile of 7 edible insects: Focus on lipid class composition and protein conversion factors" has 24 pages with 76 actual references, and 2 Figures and 5 Tables.
Almost half of the resources (30) are used in the Introduction chapter.
I have more comments about the formal editing and writing style of the article, the experiment is sufficiently described from the point of killing of the insect larvae to its analyses.
Thanks for your comments.
Abstract
To make a statement about the results, it is better to state the evidence, i.e. whether a significant difference was found or not.
Significant differences were added in the abstract in accordance with the comment.
Words from the title of the article are not usually included in the keywords, due to duplication in the search. I would like to mention the names of some representatives of the insects used in the experiment as a hint for improvement.
Thanks, the words that appeared in the title were removed and new keywords are included following your recommendation.
There are 30 references in the introductory chapter. This is not harmful if their use is not rather declarative, see lines 53 to 55, where 10 references are used without a significant purpose.
Thanks for the recommendation, the bibliography has been revised and irrelevant references have been removed. For example, see lines 54-55.
In my opinion, the material and methods chapter is more suitable for using at least the same number of citations in the text, as a reference to the fact that the analyzes were used in this case in a comparable way, as is customary.
The bibliography used in materials and methods is exactly the one that was consulted and followed for the study. The incorporation of more bibliography in this section would not be relevant in our opinion.
The results and discussions presented in the next chapter are clear, I have no comments here.
Thank you for your comment.
Reviewer 3 Report
Comments and Suggestions for Authors
Comments on Fatty acid and amino acid profile of 7 edible insects: Focus on lipid class composition and protein conversion factors
1. Line 10-12. Please explain why these seven species were chosen. What criteria were employed? Is it for nutritional benefit or for other reasons?
2. Line 22. At the end of the Abstract, please provide the applicability of this work.
3. Line 36-38. Feed composition is another factor that influences insect lipid content and fatty acid profile. Please take this into consideration.
4. Line 67-72. This reference should be added in the statement regarding the protein conversion factor due to the presence of non-protein nitrogens in insects. Chinarak, K., Panpipat, W., Summpunn, P., Panya, A., Phonsatta, N., Cheong, L. Z., & Chaijan, M. (2021). Insights into the effects of dietary supplements on the nutritional composition and growth performance of sago palm weevil (Rhynchophorus ferrugineus) larvae. Food Chemistry, 363, 130279.
5. Line 83. Please italicize all the scientific name.
6. Line 95. Why was a relatively small amount of insect powder sampled? In addition, the extract from a 100 mg insect powder was subjected to lipid class analysis. What makes it trustworthy?
7. Tables 1-4. Why a percentage below 1% was designated as tr?
8. From Table 2-4. Please discuss on the possible oxidative susceptibility or stability of each insect’s lipid. The antioxidant activity of phospholipid can be incorporated in the discussion which can be used to prevent the lipid oxidation of each insect. Also, endogenous lipid soluble antioxidants like tocopherol, carotenoids, phenolated lipids or other related species should be considered in order to fulfill the statement of lipid stability.
9. When comparing insects to standard proteins, the PDCAAS (Protein Digestibility Corrected Amino Acid Score) value of each insect should be mentioned.
Comments on the Quality of English LanguageMinor editing of English language required.
Author Response
Responses to Reviewers
First and foremost, the authors would like to thank the reviewers for taking the time to review the manuscript and make recommendations for improving its quality. Based on their input, we substantially revised the manuscript. Below, the changes made in response to the reviewer comments are presented.
Comments on Fatty acid and amino acid profile of 7 edible insects: Focus on lipid class composition and protein conversion factors
- Line 10-12. Please explain why these seven species were chosen. What criteria were employed? Is it for nutritional benefit or for other reasons?
The reason was added. See lines 84-87.
- Line 22. At the end of the Abstract, please provide the applicability of this work.
The abstract was adapted and the goal was added.
- Line 36-38. Feed composition is another factor that influences insect lipid content and fatty acid profile. Please take this into consideration.
It was added, see line 37.
- Line 67-72. This reference should be added in the statement regarding the protein conversion factor due to the presence of non-protein nitrogens in insects. Chinarak, K., Panpipat, W., Summpunn, P., Panya, A., Phonsatta, N., Cheong, L. Z., & Chaijan, M. (2021). Insights into the effects of dietary supplements on the nutritional composition and growth performance of sago palm weevil (Rhynchophorus ferrugineus) larvae. Food Chemistry, 363, 130279.
The reference was added, see line 68.
- Line 83. Please italicize all the scientific name.
It was corrected.
- Line 95. Why was a relatively small amount of insect powder sampled? In addition, the extract from a 100 mg insect powder was subjected to lipid class analysis. What makes it trustworthy?
These analyses were conducted at KU Leuven research group Food & Lipids who have ample experience in these analyses (see acknowledgement and see reference [30]). Based on their laboratory experience, it has been found that this amount of powder is sufficient to obtain reliable results, and that a larger amount could saturate the SPE columns and lead to quantification failures. For greater reliability, a high number of repetitions (n=10) was carried out.
- Tables 1-4. Why a percentage below 1% was designated as tr?
This is because it is close to the limit of quantification and is such a small percentage that it is not representative.
- From Table 2-4. Please discuss on the possible oxidative susceptibility or stability of each insect’s lipid. The antioxidant activity of phospholipid can be incorporated in the discussion which can be used to prevent the lipid oxidation of each insect. Also, endogenous lipid soluble antioxidants like tocopherol, carotenoids, phenolated lipids or other related species should be considered in order to fulfill the statement of lipid stability.
Although we considered commenting on antioxidant activity, we decided not to do so as we did not perform any analyses in this respect.
- When comparing insects to standard proteins, the PDCAAS (Protein Digestibility Corrected Amino Acid Score) value of each insect should be mentioned.
The analyses performed in this study do not allow the calculation of PDCAAS.